# Effect of Obstructive Sleep Apnea during Pregnancy on Fetal Development: Gene Expression Profile of Cord Blood

**DOI:** 10.3390/ijms25105537

**Published:** 2024-05-19

**Authors:** Laura Cànaves-Gómez, Aarne Fleischer, Josep Muncunill-Farreny, María Paloma Gimenez, Ainhoa Álvarez Ruiz De Larrinaga, Andrés Sánchez Baron, Mercedes Codina Marcet, Mónica De-La-Peña, Daniel Morell-Garcia, José Peña Zarza, Concepción Piñas Zebrian, Susana García Fernández, Alberto Alonso

**Affiliations:** 1Instituto de Investigación Sanitaria Illes Balears (IdISBa), 07120 Palma de Mallorca, Spain; laura.canaves@ssib.es (L.C.-G.); aarne.fleischer@ssib.es (A.F.); josep.muncunill@ssib.es (J.M.-F.); mariap.gimenez@ssib.es (M.P.G.); monica.delapena@ssib.es (M.D.-L.-P.); daniel.morell@ssib.es (D.M.-G.); josea.pena@ssib.es (J.P.Z.); m.c.pinas@ssib.es (C.P.Z.); susana.garcia@ssib.es (S.G.F.); 2Genomic & Bioinformatics Platform, IdISBa, 07120 Palma de Mallorca, Spain; 3Hospital Universitario de Araba, 01009 Vitoria-Gasteiz, Spain; ainoa.alvarezruizdelarrinaga@osakidetza.eus; 4Departamento de Neurociencias, Instituto de Investigación Sanitaria Bioaraba, 01009 Vitoria-Gasteiz, Spain; 5Hospital Universitario Miguel Servet, 50009 Zaragoza, Spain; andr1944@separ.es; 6Servicio de Endocrinología, Hospital Universitari Son Espases, 07120 Palma de Mallorca, Spain; mercedes.codina@ssib.es; 7Servicio de Neumología, Hospital Universitari Son Espases, 07120 Palma de Mallorca, Spain; 8Facultad de Medicina, Universidad de las Islas Baleares, 07122 Palma de Mallorca, Spain; 9Centro de Investigación Biomédica en Red de Enfermedades Respiratorias, Instituto de Salud Carlos III (CIBERES), 28029 Madrid, Spain; 10Servicio de Análisis Clínicos, Hospital Universitari Son Espases, 07120 Palma de Mallorca, Spain; 11Servicio de Pediatría, Hospital Universitari Son Espases, 07120 Palma de Mallorca, Spain

**Keywords:** obstructive sleep apnea (OSA), pregnancy, fetal development, gene expression, cord blood, qPCR, rapid eye movement (REM) sleep, perinatal outcomes, oxidative stress, hypoxia

## Abstract

Obstructive sleep apnea (OSA) is quite prevalent during pregnancy and is associated with adverse perinatal outcomes, but its potential influence on fetal development remains unclear. This study investigated maternal OSA impact on the fetus by analyzing gene expression profiles in whole cord blood (WCB). Ten women in the third trimester of pregnancy were included, five OSA and five non-OSA cases. WCB RNA expression was analyzed by microarray technology to identify differentially expressed genes (DEGs) under OSA conditions. After data normalization, 3238 genes showed significant differential expression under OSA conditions, with 2690 upregulated genes and 548 downregulated genes. Functional enrichment was conducted using gene set enrichment analysis (GSEA) applied to Gene Ontology annotations. Key biological processes involved in OSA were identified, including response to oxidative stress and hypoxia, apoptosis, insulin response and secretion, and placental development. Moreover, DEGs were confirmed through qPCR analyses in additional WCB samples (7 with OSA and 13 without OSA). This highlighted differential expression of several genes in OSA (*EGR1*, *PFN1* and *PRKAR1A*), with distinct gene expression profiles observed during rapid eye movement (REM)-OSA in pregnancy (*PFN1*, *UBA52*, *EGR1*, *STX4*, *MYC*, *JUNB,* and *MAPKAP*). These findings suggest that OSA, particularly during REM sleep, may negatively impact various biological processes during fetal development.

## 1. Introduction

Obstructive sleep apnea (OSA) is the most common sleep-related breathing disorder, and it is defined by recurrent episodes of complete (apneas) or partial (hypopneas) airflow obstruction in the upper airway while the patient sleeps [1]. Pregnancy increases the prevalence of OSA due to several physiological factors, such as weight gain, hormonal changes, or upper airway abnormalities [2,3].

OSA results in fluctuating oxygen levels, arousals, increased heart rate, blood pressure, and systemic inflammation [4], all of which can negatively impact pregnancy/fetal outcomes [5]. Accumulating evidence suggests that pregnancy-related OSA has been linked to some adverse events, including gestational diabetes mellitus (GDM), preeclampsia, preterm birth, and neonatal low weight [6,7]. However, high variability of reported rates [8,9,10,11,12] and inconsistent results have been described so far [13,14,15]. Moreover, rapid eye movement (REM) is associated with increased severity of OSA in the general population. REM-related OSA is common in young women, particularly in mild–moderate OSA, and it has been independently related with insulin resistance [16], neurocognitive symptoms [17], and cardiovascular consequences [18]. Furthermore, significant associations have been found between REM-apnea hypopnea index (AHI) with inflammatory biomarkers in pregnant women [4], higher nocturnal glucose levels [19], and increased GDM risk [20].

Although there is some information about perinatal consequences, the effects of intrauterine exposure to maternal OSA on childhood and adulthood long-term outcomes remain unknown. Fetal programming theory suggests that dysregulation in utero environment modifies the course of fetal development, resulting in long-lasting changes in the structure and function of biological systems. These may play a key role in determining lifelong health trajectories that can lead to developmental changes that may increase disease risk, like in cardiovascular and metabolic syndromes [21,22]. The specific underlying mechanisms remain elusive, but genetic and epigenetic changes may play a key role in these processes [8,23]. The hostile intrauterine environment secondary to maternal breathing pauses and nocturnal hypoxia could potentially impact the expression of certain genes, which may affect the metabolism and physiology in the most vulnerable period of life, such as the fetal period, leading to alterations in its growth patterns and to medium and long-term clinical outcomes. Maternal OSA could result in changes in birth weight (BW). Interestingly, a high BW has been found to determine the risk of obesity in school-age children [24]. In addition, an experimental study using a rat model found out that maternal hypoxia is associated with changes of periventricular white matter in adult offspring [25]. Other studies suggested that OSA during pregnancy affects the growth of head circumference and adiposity parameters [26] as well as the normal development on early childhood [27,28].

Several studies have analyzed gene expression in samples from adult patients with OSA, with the aim of identifying biomarkers for diagnosis, developing new therapeutic targets, and better understanding the pathophysiology mechanisms. These studies have found changes in the expression of genes in patients with OSA compared to controls [29,30,31]. The genes with altered expression were associated with various biological pathways and processes, including inflammation, oxidative stress, metabolism, cardiovascular function, and neurotransmission, among others [32]. Contrary to what occurs in adulthood, few studies have evaluated the influence of OSA during pregnancy or on gene expression patterns in both maternal and fetal tissues [33,34]. Some of the identified genes include those involved in inflammation (e.g., *IL-6*, *TNF-alpha*) [4]. Additionally, genes related to placental function and fetal development, such as leptin [33,35], may also play a role in mediating the effects of maternal OSA. Yet, contradictory data have also been reported [34]. Moreover, studies are still scarce and heterogeneous, and they show some limitations regarding, for example, sleep assessment procedures, sample size, and scientific designs. Therefore, many uncertainties still remain about medium- and long-term outcomes of maternal OSA on their offspring.

Some animal models have shown the health consequences of OSA and intermittent hypoxia during pregnancy on both mothers and offspring. Conducted in rats and mice, these experiments have revealed a variety of outcomes, including fetal growth restriction, hypertension in offspring, morphological and functional changes in the placenta, and long-term cardiovascular alterations in descendants. These findings underscore the importance of understanding the effects and mechanisms of poor outcomes of OSA during pregnancy on both mothers and their offspring, providing a solid foundation for future research on preventive and therapeutic interventions [36,37,38].

Gene expression studies on umbilical cord blood have provided valuable insights into the genetic underpinnings of various diseases. They encompass a wide range of conditions, such as genetic disorders, developmental abnormalities, immune system dysregulation, and metabolic diseases. These studies have unveiled intricate gene regulatory networks and identified potential biomarkers for disease prediction, diagnosis, and therapeutic interventions [8,39].

Several studies have investigated the impact of maternal OSA on maternal health and certain molecular pathways. However, to our knowledge, no previous data on gene expression in whole cord blood (WCB) from pregnant women with OSA have been published yet. For this reason, our study could provide valuable insights into understanding the comprehensive effects of OSA on both maternal and fetal health outcomes, thus opening the avenue to the development of targeted interventions to improve pregnancy outcomes in this population.

The main aim of this study was to assess the impact of maternal OSA on the fetus by studying the gene expression in WCB samples from women with and without OSA, which was confirmed during the third trimester of pregnancy through full polysomnography (PSG). As a secondary objective, we evaluated the impact of maternal OSA during REM sleep on fetal gene expression.

## 2. Results

### 2.1. Subjects of Study

A total of 10 and 20 women were included in the microarray and qPCR cohorts, respectively (Figure 1). The anthropometric characteristics of pregnant women are shown in Table 1. There were no differences in age and pregestational BMI. As expected, patients with OSA had worse apnea–hypopnea index (AHI) and desaturation index than the non-apneic women, although there were no differences in the Epworth Sleepiness Scale (ESS).

### 2.2. Gene Expression Profiling Analysis

The mRNA expression profiling data showed 3258 mRNAs differentially expressed in WCB samples from women with OSA compared to the non-OSA group. A total of 2690 mRNAs were significantly upregulated in the OSA group, and 568 mRNAs were significantly downregulated (fold change ≥ 2.0) (Figure 1). The volcano plot illustrates the whole set of differentially expressed mRNAs, with the key genes indicated in this study (FDR adjusted *p*-value < 0.05; Figure 2a). Furthermore, the heatmap provides insights into the expression and clustering of the top 50 up- and downregulated genes, with key genes indicated and further analyzed in this study (Figure 2b).

To better understand the potential pathways that might be involved in the biological processes affected in WCB with OSA, transcripts obtained from genes encoding or related to differentially expressed mRNAs were subjected to GSEA analysis to determine which pathways are enriched in genes at both ends of the list of differentially expressed mRNAs. Gene Ontology (GO) analysis was performed to determine gene and gene product features in biological processes, cellular components, and molecular activities. A total of 23 biological processes were found to be highly enriched among the associated mRNA transcripts (FDR < 0.05). The most relevant GOs after filtering by keywords were grouped into five main clusters: placenta development, response to insulin, response to oxidative stress, positive regulation of programmed cell death, and leukocyte adhesion to vascular endothelial cell (Figure 3a,b).

Through keyword KEGG analysis, we could identify 6 out of 260 significant biological processes that potentially play an important role in the fetus when exposed to maternal OSA. These were apoptosis, insulin signaling pathways, leukocyte endothelial migration, and type 1 diabetes mellitus (Figure 4), confirming the results obtained by GO analysis. The keywords were as follows: gestational diabetes mellitus, diabetes, metabolic syndrome, preeclampsia, prematurity, abnormal fetal growth, obesity, fetal, hypertension, neurodevelopmental, OSA, sleep disorder breathing, sleep apnea, sudden death, sleep, inflammation, cellular death, death, apoptosis, insulin, insulin resistance, oxidative stress, endothelial dysfunction, endothelial, sympathetic activation, placenta, placental, placental dysfunction, placental injury and dysfunction, angiogenic, angiogenesis, hypoxia, intermittent hypoxia, sleep fragmentation, and sleep arousals.

### 2.3. Gene Validation (qRT-PCR)

Based on bioinformatical analysis, 13 candidate genes that are potentially deregulated in WCB cells upon maternal OSA were selected from the gene list previously identified in the five biological processes, including 12 upregulated mRNAs (PFN1, UBA52, EGR1, STX4, MYC, JUNB, MAPKAP1, IGF2, CAT, MCL1, PPP1CB and AKT2) and 1 downregulated mRNA (PRKAR1A) (Figure 5).

We found that the qRTPCR results were consistent with the microarray data, as we found differential expression of EGR1 and PFN1 genes in OSA patients (Figure 6). Furthermore, the expression of UBA52, STX4, PFN1, MYC, MAPKAP1, JUNB, and EGR1 genes were significantly different depending on whether they came from mothers with or without OSA during REM sleep time (Figure 7). All these genes are implicated in the response to oxidative stress and hypoxia, apoptosis, placenta development, and in insulin response and secretion.

## 3. Discussion

Our study supports the notion that the adverse effects of maternal OSA extend beyond maternal health, impacting the developing fetus through altered gene expression profiles during this critical period of life.

Bioinformatics tools are becoming increasingly relevant in studying multifactorial and complex diseases. The discovery of gene expression alterations in disease is a crucial step toward a better understanding of the pathophysiology and, ultimately, a better diagnosis and treatment.

In adulthood, OSA is a complex and multifactorial condition defined by the interplay between apnea and intermittent hypoxia, along with associated genetic expression. Intermittent hypoxia, a hallmark of OSA, triggers a cascade of physiological responses, including oxidative stress and inflammation, which in turn can modulate gene expression. This dynamic process influences various cellular pathways, contributing to the pathogenesis and progression of OSA-related complications. Moreover, the environmental context adds another layer of complexity. Lifestyle factors such as smoking, obesity, and sedentary behavior exacerbate the severity of OSA, further complicating its management and outcomes [40,41,42]. Comprehending the complex interplay among genetic predisposition, intermittent hypoxia, and environmental factors is essential to fully understand the spectrum of OSA and craft personalized therapeutic interventions.

A previous study found overall similar gene expression profiles in placenta from pregnant women with OSA compared to non-OSA [34]. The mean BMI was over 40 kg/m^2^, maybe because they over-included subjects with class III obesity. Therefore, obesity and the inclusion of women with GDM and preeclampsia in the study samples might have an impact on the findings. Additionally, it cannot be ruled out that the placenta may be able to compensate for the detrimental effects of OSA on gene expression. To the best of our knowledge, we analyzed for the first time the influence of gestational OSA on WCB gene expression profiles. Approximately 3000 genes were found to be differentially expressed, suggesting that the occurrence and consequences of maternal OSA on the offspring seems to be a complex biological process involving multiple genes and steps.

Furthermore, biological function and pathway enrichment analyses showed that DEGs were mainly involved in the GO terms or KEGG pathways associated with response to oxidative stress, insulin response and signaling pathways, placenta development, response to hypoxia, cell death, and formation of endothelial barrier. These findings provide novel insights into fetal development and fetal gene expression in maternal OSA, as umbilical cord blood samples are representative of the in utero milieu of a neonate shortly before birth.

It is commonly recognized that the prevalence of OSA during pregnancy fluctuates with maternal BMI and the timing of gestation, ranging from 8.5% in subjects with normal weight to 62% in women with obesity [43]. In recent years, growing evidence has shown that OSA may negatively impact pregnancy outcomes, such as a higher risk of GDM and preeclampsia, as well as preterm delivery, low birth weight, or higher risk of maternal morbidity [6,7]. Furthermore, a few studies have shown that OSA during the REM sleep phase may be particularly relevant in the context of pregnancy, as this phase entails increased relaxation of the upper airway muscles, thereby heightening the risk of obstructive apneas. In addition, REM OSA is usually associated with longer respiratory events and higher hypoxia than OSA during non-REM sleep [44]. Also, REM OSA has been linked to higher GDM risk [20].

*PFN1* and *EGR1* were identified in this study as differentially expressed genes in WCB samples from mothers who had OSA in the third trimester of pregnancy. Interestingly, our results also show different gene expression profiles of *UBA52*, *STX4*, *PFN1*, *MYC*, *MAPKAP1*, *JUNB*, and *EGR1* in those women with REM OSA.

The expression levels of those genes have been implicated in perinatal health issues and childhood development. They were shown to play crucial roles in various cellular processes, including transcriptional regulation (*EGR1*, *JUNB*, and *MYC*), immune response (*PRF1*), protein degradation (*UBA52*), and both intracellular signaling (*MAPKAP1*) and vesicle trafficking (*STX4*) [39,45,46,47,48]. Dysregulation of these genes has been associated with adverse perinatal outcomes such as low birth weight, intrauterine growth restriction, and neonatal complications, as well as with developmental disorders in children [26,47,49]. The observed differential expression of *PFN1* and *EGR1* suggests a potential role for these genes in the pathophysiology of OSA during pregnancy and its impact on fetal development. *PFN1*, encoding profilin 1, is involved in actin dynamics and cellular processes that are crucial for proper fetal development. The dysregulation of *PFN1* expression may reflect disturbances in cellular function and signaling pathways in response to maternal OSA [46]. In consonance with the results, Liu et al. found that a reduction in sleep duration upregulated several genes, including *PFN1*, which is also a negative regulator of the migratory and elimination capabilities of cytotoxic cells, and *JUNB*, an autoimmune gene.

The main objective of this study was to assess the impact of maternal OSA on the gene expression in WCB samples. However, the sample size of the present study was certainly insufficiently powered to find differences in perinatal outcomes. In our study, the median observed birth weight was 3205 g, and none of the subjects met the criteria for low birth weight, which could limit further analysis. Small for gestational age neonates were found to have a modified expression on the *EGR1* gene network, which is involved in the regulation of cell proliferation and oxidative stress [50]. In addition, *EGR1* was positively correlated with abdominal circumference and birthweight [39]. While we did not find significant differences in perinatal outcomes between OSA and non-OSA groups, we did find an association between altered expression of *EGR1* and OSA and REM OSA in pregnant women, all of which could support the hypothesis that differential regulation of *EGR1* may contribute to increased fetal susceptibility to complications, such as low birth weight and intrauterine growth restriction in the presence of maternal OSA [39]. Taken together, all these considerations point out that the significant alterations observed in the expression levels of all these genes underscore the potential impact of maternal OSA on fetal gene regulation during this critical period of life. These findings shed light on potential molecular mechanisms underlying this relationship, and they suggest that maternal OSA, particularly during REM sleep, may influence fetal gene expression patterns associated with key cellular functions, which could contribute to the adverse developmental outcomes observed in the offspring of mothers with OSA, as well as to long-term health outcomes. Further mechanistic studies are needed to elucidate the specific roles and pathways of these genes in mediating the effects of maternal OSA on fetal development.

In the present study, we evaluated for the first time the influence of maternal OSA on the gene expression in WCB samples. The strengths of this study comprised its multicenter and prospective design, systematic sleep, and clinical phenotyping, including hospital PSG (the gold standard investigation for OSA in pregnancy) [6,51], as well as a careful adjustment for confounding factors, including BMI, maternal age, and the gestation age both at PSG and at delivery. Additionally, women with significant comorbidities, such as gestational hypertension, GDM, and preeclampsia, among others, were carefully excluded from this study to minimize confounding factors that may modulate gene expression profiles. Yet, as with any study, we should point out some limitations. First, all women in the OSA group had mild OSA and had a slight impact on DI. It is well established that mild OSA accounts for most women during pregnancy, which is associated with an increased risk of hypertensive disorders, gestational diabetes, and poor pregnancy outcomes [6,9]. Therefore, a gene expression study was relevant in these patients to better understand the mechanisms and consequences of OSA in this unique population. Nevertheless, the impact of moderate/severe OSA with greater magnitude of nocturnal hypoxemia during pregnancy has yet to be determined with further studies. Second, we included mainly Caucasian women in their healthy weight range, thus, our results may not be applied to different ethnic backgrounds and/or to women with overweight or obesity. Finally, the current sample size was not sufficiently large to incorporate many factors into our analysis, such as demographic variables, environmental influences, lifestyle factors, medical history, or other variables that might be relevant to the study. Nonetheless, there were no differences in BMI or in maternal and gestational age, and the size was sufficient to demonstrate significant differences in gene expression patterns.

## 4. Materials and Methods

### 4.1. Study Population

We performed a case-control study. Women were selected from a large study evaluating consequences of OSA during pregnancy from three Spanish university hospitals (Son Espases, Araba, and Miguel Servet). Inclusion criteria were (1) singleton pregnant women in the third trimester and (2) signed and informed consent (IC). Participants with previous OSA diagnosis, complicated pregnancies (gestational hypertension, GDM, preeclampsia, and any other significant obstetrical complication), other prior chronic diseases (diabetes mellitus, pulmonary, cardiac, or kidney diseases), and imminent delivery due to maternal–fetal disease were excluded. Two study cohorts were used—the microarray cohort for gene expression analysis [OSA (*n* = 5), non-OSA (*n* = 5)] and the real-time quantitative polymerase chain reaction (qRTPCR) cohort for confirmation study [OSA (*n* = 7), non-OSA (*n* = 13)]. To obtain the sample size pre-determined for the experimental groups, and to include 7 pregnant women classified as OSA, we required assessing for eligibility 110 pregnant women, 32 of whom were not selected (25 refused, 1 technical sleep study dropout, 1 twin pregnancy, 4 delivery before PSG, 1 change of address), and finally, we needed to perform PSG on 78 women. We selected 18 women from the remaining non-OSA group, who were homogeneous in age and body mass index (BMI) for comparative purposes. Both OSA and non-OSA women were randomly selected to perform the experiments in each of the cohorts. This study was approved by the institutional ethics committees of the hospitals and all subjects gave their written informed consent.

### 4.2. Clinical and Sleep Evaluation

All participants had their anthropometric, clinical, and sleep information collected, including the Epworth Sleepiness Scale (ESS). After being discharged from the hospital, clinical information for all included women and their infants was collected.

At recruitment, all women underwent a polysomnography (PSG) at the sleep unit. Standard criteria for epochs of 30 s were used. Thoracoabdominal stain gauges, oronasal thermistors, and nasal cannulas were used to measure breathing. A pulse oximeter was used to assess the oxyhemoglobin saturation (SaO_2_) simultaneously. Apnea was defined as the lack of airflow (<90% reduction) for at least 10 s. Hypopnea was defined as an airflow reduction (30–90%) for at least 10 s with a drop in SaO_2_ ≥ 3% or final arousal. The total number of apneas/hypopneas divided by the total number of hours of sleep defined the apnea–hypopnea index (AHI). The ratio of apneas/hypopneas during rapid eye movement (REM) sleep to the total number of hours spent in REM was used to determine the REM-AHI. OSA and REM-OSA were defined as an AHI, and REM-AHI of ≥5 h^−1^, respectively. We calculated as indicators of nocturnal SaO_2_, the mean SaO_2_ throughout the course of the night, the minimum SaO_2_ (lowest values recorded during sleep), the proportion of the study’s total time spent with SaO_2_ 90% (CT90%), and the total number of drops in SaO_2_ ≥ 3% divided by the total number of hours of sleep (desaturation index (DI)).

### 4.3. Sample Acquisition and Preparation

WCB samples were collected after delivery. Total RNA was isolated with Qiagen RiboPure kit TRI reagent-based protocol (Merck, Darmstadt, Germany) from blood cells using 250 µL of WCB and finally eluted in RNAse-free water. RNA concentration and purity was assessed measuring absorbance at 260 nm and 280 nm using a Synergy H1 spectrophotometer (BioTek Instruments, Santa Clara, CA, USA).

### 4.4. Gene Expression Analysis

Genome-wide gene expression analysis was performed at the IdISBa Genomics Unit (Palma, Spain) using human Clariom D microarrays (Thermo Fisher Scientific, Waltham, MA, USA) on total RNA extracted from cord blood cells from 10 neonates (five from the OSA group and five from the non-OSA group). Sample preparation was accomplished in accordance with the instructions detailed in the GeneChip WT PLUS reagent kit (Thermo Fisher Scientific, Waltham, MA, USA). Briefly, 90 ng of total RNA aliquots was used from each sample to achieve, via reverse transcription, second strand cDNA synthesis and in vitro transcription of 15 µg cRNA per sample. Sense-strand dUTP-labeled cDNA probes were synthesized through reverse transcription of cRNA, followed by cRNA hydrolysis. 5.2 μg of fragmented biotin-labeled cDNA probes, prepared in 160 μL hybridization cocktail, were hybridized to human Clariom D microarrays for 16 h at 45 °C with a rotation of 60 rpm. Subsequently, microarrays were washed and stained with streptavidin–phycoerythrin using the GeneChip Fluidics Station 450 (Thermo Fisher Scientific, Waltham, MA, USA) and scanned at 0.7 μm resolution using the GeneChip Scanner 3000 7G (Thermo Fisher Scientific, Waltham, MA, USA).

Background correction, normalization, and summarization of microarrays was conducted using the oligo package (version 1.66, Bioconductor). Quality control for the data was performed using the arrayQualityMetrics package (version 3.55, Bioconductor).

Differentially expressed genes (DEG) were identified using the limma package (version 3.58, Bioconductor), considering genes with an absolute fold change cut-off of >2.0 and an adjusted false discovery rate (FDR) *p*-value < 0.05. The data were deposited in the GEO database under the accession number GSE264558.

Functional enrichment analysis was performed using the Bioconductor packages Clusterprofiler (version 4.11) and fgsea (version 1.28) for Gene Ontology (GO) [52], Kyoto Encyclopedia of Genes and Genomes (KEGG) [53], and WikiPathways [54] databases. For each database, over-representation analysis (ORA) and gene set analysis enrichment (GSEA) were conducted. All analyses were performed using R (version 4.3.2).

### 4.5. qRTPCR Validation

qRT-PCR was conducted on the samples from the two groups (7 in the OSA group and 13 in the non-OSA group) to confirm a subset of DEG identified in the microarray study. First, RNA reverse transcription was performed using the SensiFAST cDNA Synthesis kit protocol (Bioline, London, UK) to create individual cDNA libraries from 1 μg of each total RNA sample. qRT-PCR was performed using the qRT-PCR system CFX96 Touch (Bio-Rad, Hercules, CA, USA). Then, 2 μL of 1/10 diluted cDNA was mixed with 5 μL of Qiagen AMPLIFYME SG Universal Mix (Merck, Darmstadt, Germany), 0.2 μL of each forward and reverse primer and with DNAse/RNAse free water, up to a final volume of 10 μL per well. The primers used in this study are listed in Table 2.

### 4.6. Statistical Analysis

The anthropometric characteristics and sleep study data of the women are shown in the table. Laboratory data of subjects included in the microarray, and the qPCR cohorts are presented as median + interquartile range (IQR) or percentage (Table 1). Differences between groups were analyzed using Student’s t-test or Mann–Whitney U test for continuous variables, and Fisher’s exact test (two-tailed) or chi-squared test were used for categorical variables. Gene data analysis was performed using RStudio [55]. The relative expression of messenger RNA (mRNA) was determined by the 2^−∆∆Ct^ method normalized to 18S rRNA expression. Chi-test was applied to verify the statistical significance of mRNA expression levels between patients and controls. Error bars in graphs represent the standard error of the mean (SEM). Median with interquartile range is represented in the scatter plots.

## 5. Conclusions

In conclusion, WCB cells showed distinct gene expression profiles depending on the presence of maternal OSA and OSA during REM during pregnancy. This could potentially have an adverse influence on several biological and molecular processes throughout fetal development, which could finally contribute to long-term adverse health outcomes in the offspring. However, these promising findings need to be further validated in larger studies to establish a solid molecular foundation for the understanding of the adverse effects of maternal OSA on fetal health, which could finally lead to the development of therapeutic interventions for better pregnancy outcomes in this population.

## Figures and Tables

**Figure 1 ijms-25-05537-f001:**
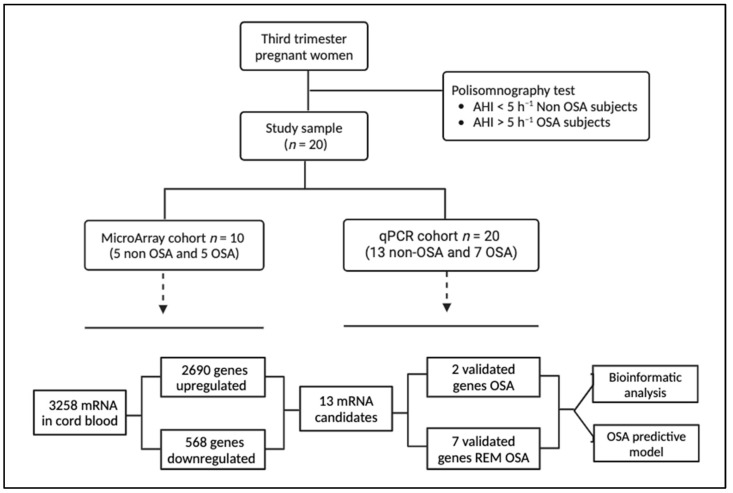
Flowchart of the study. Pregnant women in the third trimester were recruited, and after a PSG, they were divided for the study on the basis of non-OSA and OSA and further divided into the microarray cohort and the qPCR cohort. Five non-OSA and five OSA patients were used to perform a general screening using microarrays. A total of 13 out of 3258 differentially expressed mRNAs were validated in an independent cohort by qRT-PCR.

**Figure 2 ijms-25-05537-f002:**
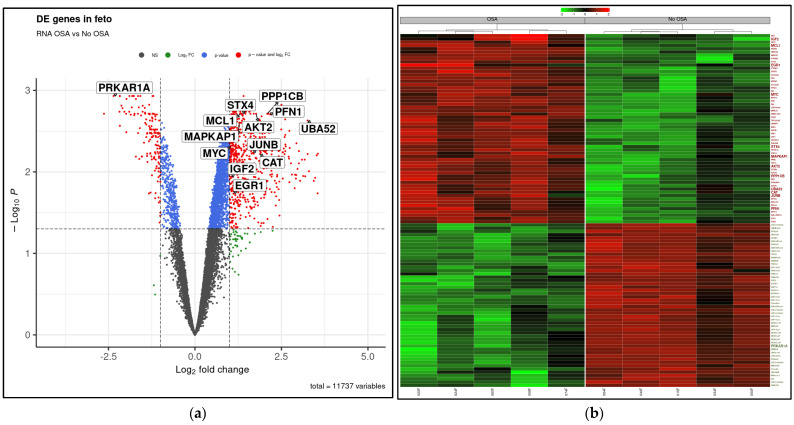
WCB differential gene expression in OSA compared to non-OSA. (**a**) Volcano plot showing significance (−log_10_ transformed *p* values) and magnitude (log_2_(fold change)) of differential gene expression in WCB from OSA compared to non OSA with *p*-value ≤ 0.05 and fold change ≥ 2.0 filters. Red dots showing significant DE genes significantly for both filters. (**b**) Heat map illustrating magnitude of differential gene expression in WCB from OSA compared to non-OSA. Green and red color represents the top 50 of under- and overexpressed genes, respectively.

**Figure 3 ijms-25-05537-f003:**
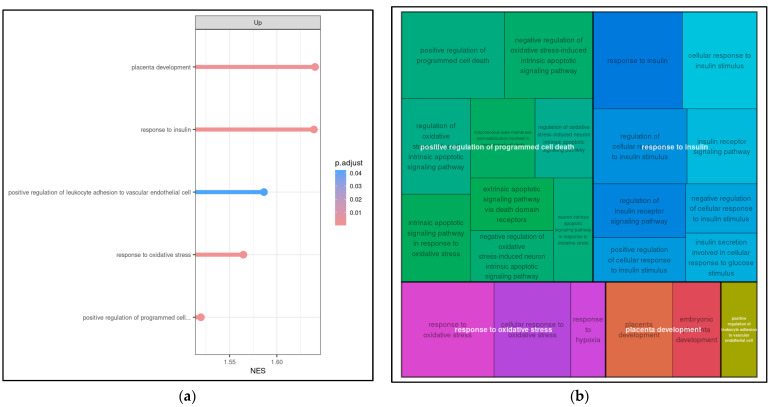
Potential biological pathways altered in WCB cells from OSA-affected mothers identified by microarray analysis. (**a**) Enrichment analysis of gene sets in the GO (Gene Ontology) database (GO-GSEA). For functional enrichment, “keywords” (pathologies and signaling pathways) associated with OSA were introduced. Significant positive NES (normalized enrichment scores) value indicates that members of a given gene set tend to appear at the top of the ranked transcriptome data. (**b**) The analysis after filtering by keywords showed 23 entries represented by the small squares, which were grouped into five main clusters: placenta development, response to insulin, response to oxidative stress, positive regulation of programmed cell death, and leukocyte adhesion to vascular endothelial cell.

**Figure 4 ijms-25-05537-f004:**
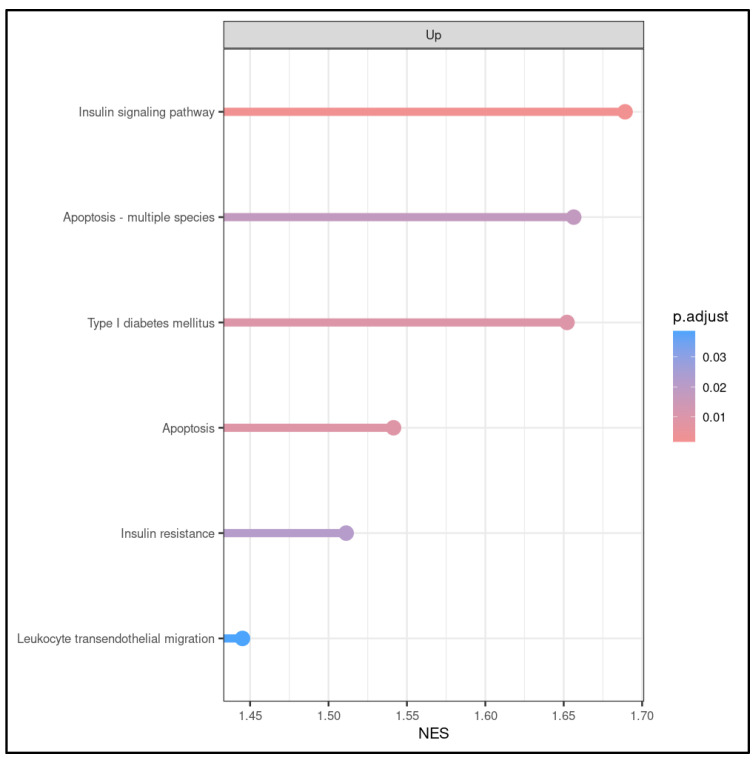
KEGG (KEGG-GSEA) pathway enrichment analysis reveals dysregulated pathways in WCB cells with OSA compared to control samples. KEGG pathway analysis led to the identification of 260 significant biological processes, among which, six of them were related to apoptosis, insulin resistance and signaling pathway, leukocyte endothelial migration, and type 1 diabetes mellitus.

**Figure 5 ijms-25-05537-f005:**
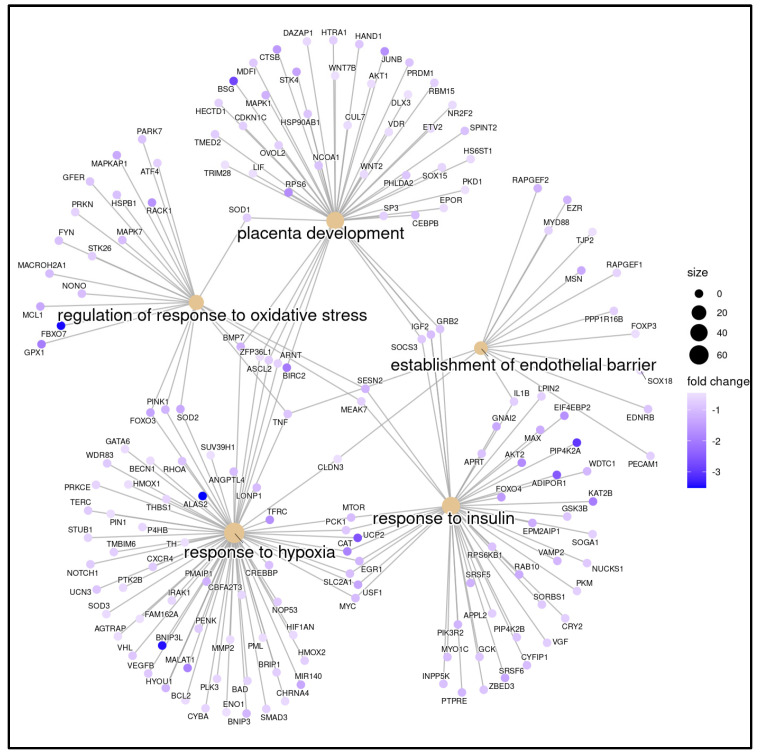
Potential signaling pathways triggered in WCB cells by maternal OSA. Pathway analysis indicated that five pathways corresponded to upregulated transcripts. The most enriched network was “response to oxidative stress”, comprising four target genes. The intensity of the blue color corresponds to the fold change; the higher the intensity, the higher the value. The size of the circle depends on the number of genes associated with each network.

**Figure 6 ijms-25-05537-f006:**
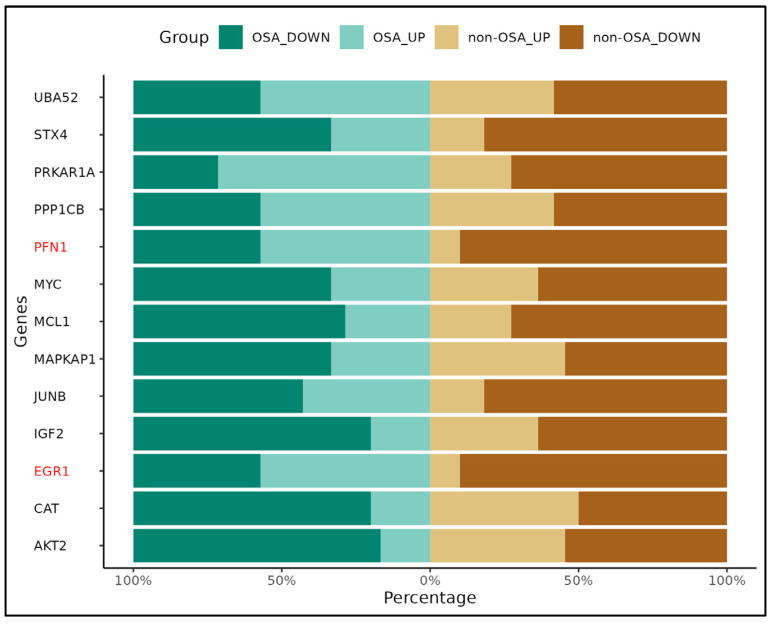
Gene expression patterns obtained by qRT-PCR in WCB with OSA versus non-OSA. Genes are represented on the *Y*-axis, while the *X*-axis denotes the frequency of their expression. Horizontal bars are colored in green for OSA samples and brown for non-OSA samples. Genes with significant differential gene expression pattern are highlighted in red. The different shades of green or brown indicate whether genes are up- or downregulated in WCB cells affected or not affected by maternal OSA.

**Figure 7 ijms-25-05537-f007:**
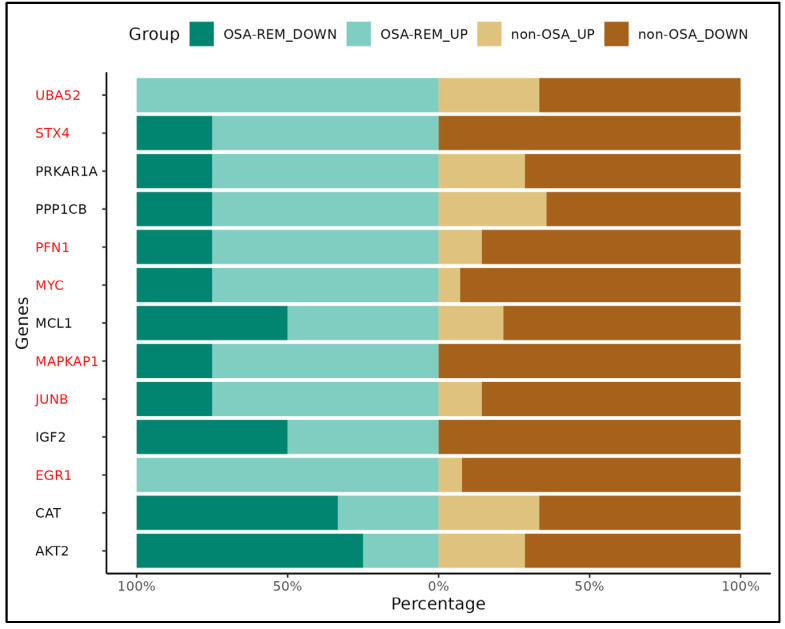
Gene expression patterns obtained by qRT-PCR in WCB with REM OSA versus non-REM OSA. Genes are represented on the *Y*-axis, while the *X*-axis denotes the frequency of their expression. Horizontal bars are colored in green for REM OSA samples and brown for non-REM OSA. Genes with significant differential gene expression pattern are highlighted in red. The different shades of green or brown indicate whether genes are up- or downregulated in REM OSA compared to non-REM OSA samples.

**Table 1 ijms-25-05537-t001:** Anthropometric characteristics sleep and laboratory data of subjects included in the microarray and the qRT-PCR cohorts.

	Microarray Cohort			qRT-PCR Cohort		
	OSA (*n* = 5)	Non-OSA (*n* = 5)	*p*	OSA (*n* = 7)	Non OSA (*n* = 13)	*p*
Age (years)	36 (36–40)	36 (33–37)	0.42	36 (34–38)	34 (33–37)	0.64
BMI before pregnancy (kg/m^2^)	22.3 (21–24)	21.9 (20.1–21.9)	0.73	22.3 (20.8–24.8)	21.7 (20.8–22.7)	0.64
Neck circumference (cm)	34 (32–34.5)	32 (34–37)	0.53	34 (31.8–34.8)	33.5 (32–34)	0.94
Waist (cm)	95.5 (95–106)	107 (96–112)	0.35	99.5 (95.3–106.3)	104 (96–108)	0.49
Hip circumference (cm)	102 (99.25–104)	107.5 (99–116)	0.56	102 (99.3–104)	106.5 (97–115)	0.86
Gestational age at PSG (weeks)	36 (32.5–38)	31 (30–33)	0.11	34 (31–37)	34 (31–35)	0.77
First pregnancy (n)	3	1	0.52	4	5	0.64
Gestational smokers (n)	1	0	1	1	0	0.35
Epworth Sleepiness Scale	7 (6–9)	4 (3–5)	0.1	7 (5–8)	6 (4–10)	1
AHI (h^−1^)	6.4 (5.3–7.6)	0.1 (0.0–0.2)	0.01	6.4 (5.5–8.6)	0.4 (0.1–0.7)	0.00
Desaturation index (h^−1^)	0.3 (0–1.1)	0 (0–0.1)	0.05	0.6 (0.3–1.7)	0 (0–0.4)	0.01
REM-AHI (h^−1^)	10 (4.8–10.3)	0 (0–1)	0.05	10 (3.3–16.5)	0 (0–1)	0.00
Birthweight (g)	3180 (3015–3270)	3210 (3200–3225)	0.84	3180 (2967–3265)	3232 (3080–3430)	0.40
Cesarean section (n)	1	1	1	1	1	1
Neonatal gestational age (weeks)	39 (38–40)	39 (38–40)	1	39 (39–40)	40 (38–40)	0.39

Descriptive analysis and comparison between groups. Continuous variables are presented as median + interquartile range (IQR) or percentage. Abbreviations: PSG, polysomnography; AHI, apnea-hypopnea index; BMI, body mass index; qRTPCR, real-time quantitative polymerase chain reaction; cm, centimeter; g; grams; h, hours; h^−1^, per hour.

**Table 2 ijms-25-05537-t002:** Primers (KiCqStart^®^, Sigma-Merck, Darmstadt, Germany) used to confirm differentially expressed genes by qRTPCR.

Selection of Primers	Reference Sequence ID	Exon Boundary	Forward	Reverse
*18S rRNA*	NR_145820.1		5′-TAAGCAACGAGACTCTGGCAT	5′-CGGACATCTAAGGGCATCACAG
*AKT2*	NM_001243027	2–3	5′-ACTTCCTGCTGAAGAGCGAC	5′-CCTCCCTCTCGTCTGGAGAA
*CAT*	NM_001752	12–13	5′-CTCTTCTGGACAAGTACAATG	5′-AGGAGAATCTTCATCCAGTG
*EGR1*	NM_001964	1–2	5′-GCAGAGTCTTTTCCTGAC	5′-TTGGTCATGCTCACTAGG
*IGF2*	NM_000612	3–4	5′-GGACAACTTCCCCAGATAC	5′-GTGGGTAGAGCAATCAGG
*JUNB*	NM_002229	NA	5′-TACTGTGGAAAAGAAACACG	5′-GAACAAACACACACAAACAC
*MAPKAP1*	NM_001006617	6–7	5′-TAAATGCTGCTCATGGATTC	5′-AACTTTCTGGGATCCTTTTC
*MCL1*	NM_001197320	3–4	5′-TAGTTAAACAAAGAGGCTGG	5′-ATAAACTGGTTTTGGTGGTG
*MYC*	NM_002467	2–3	5′-TGAGGAGGAACAAGAAGATG	5′-ATCCAGACTCTGACCTTTTG
*PFN1*	NM_005022	2–3	5′-TTCTTGTTGATCAAACCACC	5′-GGGAATTTAGCATGGATCTTC
*PPP1CB*	NM_002709	NA	5′-GACCATAGCAAATCACAGAG	5′-GCAATCTATGGAGCAGATTC
*PRKAR1A*	NM_002734	6–7	5′-GAAGGGGATAACTTCTATGT	5′-ATTCTTCTATAGCTGTCTCGG
*STX4*	NM_004604	9–10	5′-GATTGAGAAGAACATCCTGAG	5′-CATTATCCAACCACTGTGAC
*UBA* *52*	NM_001033930	4–5	5′-CCTTATTTGACCTTCTTCTTGG	5′-CTTGCCCAGAAATACAACTG

## Data Availability

https://www.ncbi.nlm.nih.gov/geo/query/acc.cgi?acc=GSE264558 (accessed on 25 April 2024).

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
