# Peer review of "Effect of Obstructive Sleep Apnea during Pregnancy on Fetal Development: Gene Expression Profile of Cord Blood"

_ijms, 2024, doi:10.3390/ijms25105537_

Round 1

Reviewer 1 Report

Comments and Suggestions for Authors

The authors have written a well-documented research article on gene expression in fetal cord blood from mothers diagnosed with OSA, compared to those without OSA. It is an interesting subject and a novel approach that might be of interest to the medical community. Here are some remarks and suggestions for improving your paper form:

1. The authors have discussed the importance and innovation of conducting their study in detail in the introduction section. However, I suggest that the final paragraph of the introduction should focus on better outlining the research question and the primary and secondary objectives of the study. For instance, the primary objective could be to investigate the differences between genes resulting from mothers with and without obstructive sleep apnea (OSA), while one secondary objective could be to observe the impact of OSA during REM sleep or other factors mentioned in the results section. This approach will provide a clearer and more coherent overview of the findings. Also, the sentence" to examine the effect of diverse OSA criteria"(line 109) is somehow incomplete and needs some additional explanations.

2. I suggest that you mention the type of article in the methodology section and provide details about the selection method used for participants. It would be helpful to know the duration of the selection process and how pregnant women were recruited. Additionally, it is beneficial to mention the inclusion criteria. Furthermore, I suggest providing more information about the two cohorts used, including how the patients were divided or chosen for each cohort, and also explaining the number of patients in each cohort.

3. In the table of the results section, the authors have included various paraclinical findings such as creatinine and cholesterol levels. I am wondering why this specific data has been provided and what purpose it serves in the analysis. It would be helpful if the authors could explain the significance of these findings and how they contribute to the overall analysis.

4. The participants were divided into two groups based on their Apnea-Hypopnea Index (AHI): those with Obstructive Sleep Apnea (OSA) and those without. However, even though the median AHI value falls into the OSA category, it is still considered to be in the lower range of mild OSA. Additionally, the Oxygen Desaturation Index (ODI), which is an important parameter for determining the severity of OSA, has normal values in the table. Therefore, the authors should consider this while interpreting their results. 

Comments on the Quality of English Language

A few minor modifications need to be made. Firstly, on line 64, "increase decrease risk" should be revised for clarity.

Secondly, the paragraph from the introduction section, specifically from lines 100 to 105, is too long and difficult to follow. To address this, it is suggested that the phrases be separated into shorter, more coherent sentences to make the statements clearer and more understandable.

Reviewer 2 Report

Comments and Suggestions for Authors

enclosed comments
